# Aging decreases docosahexaenoic acid transport across the blood-brain barrier in C57BL/6J mice

**Takuro Iwao, Fuyuko Takata, Junichi Matsumoto, Hisataka Aridome, Miho Yasunaga, Miki Yokoya, Yasufumi Kataoka, Shinya Dohgu** *

Department of Pharmaceutical Care and Health Sciences, Faculty of Pharmaceutical Sciences, Fukuoka University, Jonan-ku, Fukuoka, Japan

* dohgu@fukuoka-u.ac.jp

**Data Availability Statement:** All relevant data are within the paper and its Supporting information files.

## Abstract

Nutrients are actively taken up by the brain via various transporters at the blood–brain barrier (BBB). A lack of specific nutrients in the aged brain, including decreased levels of docosahexaenoic acid (DHA), is associated with memory and cognitive dysfunction. To compensate for decreased brain DHA, orally supplied DHA must be transported from the circulating blood to the brain across the BBB through transport carriers, including major facilitator superfamily domain-containing protein 2a (MFSD2A) and fatty acid-binding protein 5 (FABP5) that transport esterified and non-esterified DHA, respectively. Although it is known that the integrity of the BBB is altered during aging, the impact of aging on DHA transport across the BBB has not been fully elucidated. We used 2-, 8-, 12-, and 24-month-old male C57BL/6 mice to evaluate brain uptake of [$^{14}$C]DHA, as the non-esterified form, using an *in situ* transcardiac brain perfusion technique. Primary culture of rat brain endothelial cells (RBECs) was used to evaluate the effect of siRNA-mediated MFSD2A knockdown on cellular uptake of [$^{14}$C]DHA. We observed that the 12- and 24-month-old mice exhibited significant reductions in brain uptake of [$^{14}$C]DHA and decreased MFSD2A protein expression in the brain microvasculature compared with that of the 2-month-old mice; nevertheless, FABP5 protein expression was up-regulated with age. Brain uptake of [$^{14}$C]DHA was inhibited by excess unlabeled DHA in 2-month-old mice. Transfection of MFSD2A siRNA into RBECs decreased the MFSD2A protein expression levels by 30% and reduced cellular uptake of [$^{14}$C]DHA by 20%. These results suggest that MFSD2A is involved in non-esterified DHA transport at the BBB. Therefore, the decreased DHA transport across the BBB that occurs with aging could be due to age-related down-regulation of MFSD2A rather than FABP5.

## Introduction

Many essential nutrients are necessary to maintain physiological functions. A lack of essential nutrients can cause general disorders of the central nervous system (CNS), and proper functioning of the CNS requires adequate levels of essential nutrients in the brain. Maintaining

**Funding:** This work was supported in part by: Grants-in-Aid for Scientific Research (KAKENHI; grant numbers JP20K16065 to TI, JP19K07338, and JP21H04863 to SD and JP21K06813 to FT) from the Japan Society for the Promotion of Science (https://www.jsps.go.jp/); funds to FT from the Foundation for Dietary Scientific Research (https://z-ssk.org/); funds to SD (grant numbers 196004 and 197012), IT (211032) and FT (221043) from the Central Research Institute of Fukuoka University (http://www.suisin.fukuoka-u.ac.jp/home1/); and funds to FT from the Fukuoka University Program to support the research activities of female researchers (http://www.suisin.fukuoka-u.ac.jp/jyosei/). The funders had no role in study design, data collection and analysis, decision to publish, or preparation of the manuscript.

**Competing interests:** The authors have declared that no competing interests exist.

the CNS microenvironment requires transporting various nutrients across the blood-brain barrier (BBB). The BBB comprises brain microvessel endothelial cells (BMECs) sealed with tight junctions; it restricts the paracellular transport of substances in the blood and selectively transports essential nutrients into the CNS via various specialized transporters located on BMECs [1].

It is increasingly recognized that the nutritional needs of the CNS vary in different diseases. In addition, the BBB is likely to be compromised under chronic inflammation, stroke, hypoxia, Alzheimer's disease (AD), and other neurological diseases [2], which can cause undernutrition of the brain [3]. These pathologies, as well as natural aging, induce dysfunction of the transporters and receptors and the loosening of tight junctions as well as other junctions at the BBB [4]. Indeed, chronic inflammation can lead to differential regulation of nutrient transporters [5], which contributes to the pathology of chronic inflammation-associated cognitive decline [6]. Therefore, low levels of several nutrients in the brain are associated with cognitive decline, likely due to changes in the functions of nutrient transporters at the BBB.

Docosahexaenoic acid (DHA) (22:6n-3), an n-3 polyunsaturated fatty acid (MW: 328), has attracted much attention because of its functional and structural importance in the brain. DHA is highly enriched in brain phospholipids and contributes to proper brain development and function. Moreover, many neurophysiological functions of DHA have been identified, including regulation of cell survival, neuroinflammation, neurogenesis, and participation in signal transduction [7]. It is controversial to state that the human brain's capacity to biosynthesize DHA from its precursor, α-linolenic acid (ALA, 18:3n-3), is very low [8], because maternal dietary ALA given during pregnancy-lactation increased brain DHA levels in the offspring [9]. However, plasma DHA, which is obtained directly from dietary intake to maintain DHA levels in the brain, must be transported from the blood to the brain across the BBB [10].

Brain transport mechanisms of DHA vary based on its carrier form, non-esterified or esterified DHA [7]. Non-esterified DHA (NE-DHA) is mainly found in the blood as a complex with albumin [11]. NE-DHA disassociated from albumin is taken up into the brain through passive diffusion [12]. In brain uptake of NE-DHA, fatty acid-binding protein 5 (FABP5) is directly involved in the intracellular trafficking of NE-DHA, which penetrates the luminal membrane of BMECs by passive diffusion, across BMECs [13, 14]. Esterified DHA largely exists as phospholipids, in part triacylglycerol and cholesteryl ester pools [15]. Lysophosphatidylcholine-DHA (LPC-DHA) in circulating esterified fatty acid pools is bound to albumin or is sequestered within the phospholipid membrane of lipoproteins [15]. A previous study reported that the major transporter for LPC-DHA uptake by the brain is the major facilitator superfamily domain-containing protein 2a (MFSD2A) [16], which is exclusively expressed in BMECs. That study indicated that MFSD2A did not recognize DHA as a substrate because MFSD2A transported lysophosphatidylethanolamine, LPC-oleate and LPC-palmitate [16]. It remains unclear whether NE-DHA serves as a substrate for MFSD2a, although NE-DHA is the major plasma pool supplying the brain under normal physiological conditions [17]. Therefore, it is important to investigate the brain uptake mechanisms of NE-DHA and the effect of alteration of the milieu interieur on NE-DHA transport across the BBB.

Brain DHA levels decrease with age, and this reduced DHA is associated with age-related cognitive dysfunction [18]. DHA supplementation, which elevates NE-DHA levels in plasma [19], is likely to be beneficial when administered before or during the earliest stages of cognitive decline [20]. However, the mechanisms underlying the reduction of brain DHA levels with age are not well understood. Animal and clinical studies have evidenced age-related BBB dysfunction [21, 22]; however, the impact of aging on brain DHA uptake at the BBB is currently unknown. In this study, we investigated age-related changes in NE-DHA transport

across the BBB and expression of DHA carrier proteins in mice aged 2, 8, 12, and 24 months. We further examined whether brain uptake of [14C]DHA is mediated by a transport carrier, such as MFSD2A, both, in vivo by transcardiac brain perfusion with excess DHA using 2-month-old mice and in vitro using BMECs with siRNA-mediated gene silencing.

## Materials and methods

### Animals

All protocols involving experimental animals were approved by the Laboratory Animal Care and Use Committee of Fukuoka University (permit number: 2004001, 2204002). The 2- (young), 8- (adult), 12- (middle-aged), and 24 (aged)-months old male C57BL/6J mice and Wistar rats at 3 weeks old were purchased from Charles River Laboratory (Kanagawa, Japan) and Japan SLC, Inc. (Shizuoka, Japan). Mice and rats were housed under a controlled temperature (22 ± 2°C) and light-dark cycle (lights on from 7:00 to 19:00), with access to water and chow diet *ad libitum*. After habituation for 1 week, the mice underwent the experiments.

### Measurement of brain uptake of [14C]DHA, [3H]Mannitol, and [14C] Sucrose

Brain uptake of DHA was assessed in mice using an *in situ* transcardiac brain perfusion technique. This technique was previously used by Banks et al. [23]. The [1-14C]DHA as NE-DHA (American Radiolabeled Chemicals, St. Louis, MO, USA; ARC0380), [3H]mannitol (PerkinElmer, Waltham, MA, USA, NET101) and [14C]sucrose (PerkinElmer, Waltham, MA, USA; NEC100X) were diluted to concentrations of 0.1 µCi/mL (DHA and sucrose) and 0.2 µCi/mL (mannitol) in a physiological buffer containing 141 mM NaCl (Sigma, St. Louis, MO, USA; 28–2270–5), 4 mM KCl (Wako, Osaka, Japan; 163–03545), 2.8 mM $CaCl_2$ (Sigma; 05–0580), 1 mM $MgSO_4 \cdot 7H_2O$ (Kishida Chemical Co., Ltd., Osaka, Japan; 000–46905), 1 mM $NaH_2PO_4 \cdot 2H_2O$ (Wako; 192–02815), 10 mM d-glucose (Wako; 041–00595), and 10 mM 4-(2-hydroxyethyl)-1-piperazineethanesulfonic acid (HEPES; Sigma; H4034); pH 7.4. In a competition assay, unlabeled NE-DHA (Sigma; D2534-100MG) was added to the perfusate (final concentration of unlabeled DHA: 100 µM) before infusion. Ethanol (Nacalai Tesque, Kyoto, Japan; 14713–95) was added to an equal volume of unlabeled DHA to the perfusate (final ethanol concentration: 0.001%) as the vehicle control.

Mice were anesthetized using an intraperitoneal injection of 25% urethane (Sigma; 94300). The heart was exposed, the left jugular vein was severed, and the descending aorta was ligated. Perfusate containing [14C]DHA, [3H]mannitol, or [14C]sucrose was then infused into the left ventricle of the heart at a rate of 2 mL/min for 0.5 to 1.5 min using a 27 gauge butterfly needle. In the uptake experiment for aged mice, the perfusate was infused for 1 min. After perfusion, the brain was removed, dissected into six regions (the olfactory bulb, forebrain, cortex, hippocampus, thalamus and hypothalamus, and cerebellum), and weighed. Each brain region was mixed with 1 mL of tissue solubilizer (Solvable™; PerkinElmer; 6NE9100) and incubated at 60°C for 24 h. Samples were prepared for scintillation counting by the addition of 0.2 mL $H_2O_2$ (Sigma; 13-1910-5) and 10 mL of liquid scintillation cocktail (Pico-Fluor Plus; PerkinElmer; 6013699). The radioactivity in the samples was then measured using a liquid scintillation counter (Packard 2250CA; PerkinElmer). Brain/perfusate ratios were calculated by dividing the radioactivity in 1 g of brain tissue by the radioactivity in an mL of perfusate. Whole brain values were calculated by dividing the total radioactivity in each brain region by the total weight of each brain region. The unidirectional influx rate (Ki) was determined by the linear portion of the slope in the plot of brain/perfusate ratio against perfusion time.

## Isolation of brain microvessels

Brian microvessels were isolated by the modified method of Yousif et al. [24]. Male mice aged 2, 8, 12, and 24 months were anesthetized before Dulbecco's phosphate-buffered saline (-) (D-PBS) (Wako; 045–29795;10 mL per mouse) was infused into the left ventricle of the heart. The brain tissue was triturated using a glass homogenizer coated with 1% bovine serum albumin (BSA)-Hanks' Balanced Salt Solution (HBSS; Thermo Fisher Scientific, Waltham, MA, USA; 14185–045) in 1 mL of Buffer A (HBSS containing 1% Phosphatase Inhibitor Cocktail [ethylenediaminetetraacetic acid (EDTA) free; Nacalai Tesque; 07575–51], 1% Protease Inhibitor Cocktail for use with mammalian and tissue extracts [Nacalai Tesque; 25955–11], 1 mM phenylmethylsulfonyl fluoride [PMSF; Sigma; P7626], and 15 μg/mL deoxyribonuclease I [Sigma; D4513]) on ice. This suspension was transferred into a 1.5 mL tube and centrifuged at $1,000 \times g$ for 10 min at 4°C. Next, the supernatant was aspirated and the pellet was mixed with 1 mL of 17.5% dextran (Sigma; D8821)-HBSS. The suspension was then centrifuged at $4,400 \times g$ for 15 min at 4°C. The supernatant with a lipid layer was removed and the pellet was resuspended in Buffer B (Buffer A containing 1% BSA). Next, the suspension was filtered using a 10 μm nylon mesh membrane to trap the residue, including microvessels, on the surface of the membrane. HBSS was then passed through the membrane to remove debris in the residue. The microvessels on the membrane were then washed into a 1.5 mL tube with Buffer A and centrifuged at $20,000 \times g$ for 5 min at 4°C. Finally, microvessels were obtained at the bottom of the tube and stored at -80°C until use. We confirmed that the obtained brain microvessels were enriched with brain endothelial cell-specific proteins (occludin and claudin-5).

## Extraction of total protein from brain microvessels

The microvessels were homogenized in phosphoprotein lysis buffer containing 10 mM Tris-HCl (pH 6.8; Nacalai Tesque; 35434–34), 100 mM NaCl (Sigma; 28-2270-5), 1 mM EDTA (pH 8.0; Wako; 311–90075), 1 mM EGTA (Wako; 346–01312), 10% glycerol, 1% Triton-X100 (Sigma; X100), 0.1% sodium dodecyl sulfate (SDS; Nacalai Tesque; 02873–75), 0.5% sodium deoxycholate (Sigma; D6750), 20 mM sodium pyrophosphate (Sigma; S6422), 2 mM sodium orthovanadate (Sigma; S6508), 1 mM sodium fluoride (Wako; 196–01975), 1% protease inhibitor cocktail (Sigma; P2714), 1% phosphatase inhibitor cocktail 2 (Sigma; P5726), 1% Phosphatase Inhibitor Cocktail 3 (Sigma; P0044), and 1 mM PMSF (Sigma) using an electric mixer, and then sonicated on ice. Samples were centrifuged at $15,000 \times g$ for 15 min at 4°C, and the supernatants were collected. The total protein concentration in the lysates obtained from microvessels was determined using a Pierce™ BCA Protein Assay Kit (Thermo Fisher Scientific; 23225).

## Primary rat brain endothelial cell culture

The method of primary culture of rat brain endothelial cells (RBECs) is previously described [25]. The meninges and white matter were carefully removed from the forebrains, and the gray matter was minced using a scalpel and enzymatic digestion by DMEM (Wako; 048–29763), including collagenase type 2 (1 mg/mL, Worthington, Lakewood, NJ, USA; CLS2) for 75 min at 37°C with agitation in the water bath. After inactivation by adding cold DMEM, the suspension was centrifuged ($1,000 \times g$, 8 min). The pellet was separated by centrifugation in 20% BSA, (Wako, 011–27055)—DMEM ($1,000 \times g$, 20 min). The pellet containing microvessels were further digested with DMEM including collagenase/dispase (1 mg/mL, Roche, Mannheim, Germany; 11097113001) for 20 min at 37°C with agitation in the water bath. After inactivation, microvessel clusters were separated on a 33% continuous Percoll (GE Healthcare, Buckinghamshire, UK; 17-5445-01) gradient, collected and plated on collagen type IV (0.1

mg/mL, Sigma; C5533) and fibronectin (0.075 mg/mL, Sigma; F1141-5MG) coated dishes. RBEC cultures were maintained in RBEC medium [DMEM/F12 (Wako; 042–30555) supplemented with 10% FBS (Biosera, Kansas, MO, USA; FB-1365/500), basic fibroblast growth factor (1.5 ng/mL, R&D, Minneapolis, MN, USA; 2099-FB-025), heparin (100 μg/mL, Sigma; H3149), insulin (5 μg/mL), transferrin (5 μg/mL), sodium selenite (5 ng/mL; insulin-transferrin-sodium selenite media supplement, Sigma; I1884), penicillin (100 units/mL), streptomycin (100 μg/mL; penicillin-streptomycin mixed solution, Nacalai Tesque; 09367–34) and gentamicin (50 μg/mL, Biowest, Riverside, MO, USA; L0012)] containing puromycin (4 μg/mL, Nacalai Tesque; 14861–84) at 37˚C in a humidified atmosphere of 5% $CO_2$/95% air, for three days and typically reached 70–80% confluency. RBECs were passaged to 35 mm dishes ($30 \times 10^4$ cells/dish) and 24-well plates ($10 \times 10^4$ cells/dish) and maintained in RBEC medium supplemented with 500 nM hydrocortisone (Sigma; H0135).

## siRNA transfection

RBECs cultured on a 35-mm dish were transfected with the lipid complex, including Lipofectamine® RNAiMAX Transfection Reagent (4 μL; Invitrogen, 13778075) and Rat Mfsd2a Silencer® Select Pre-designed siRNA (50 nM; Life technologies, s151458) or Silencer Select Negative Control (50 nM; Life technologies, 4390843) in RBEC medium for 2 days. The MFSD2A protein levels in RBECs were assessed using Western blot and siMfsd2a-transfected RBECs were used for DHA cellular uptake assay.

## Cellular uptake of [$^{14}$C]DHA

To measure the cellular uptake of DHA, RBECs cultured on a 24-well plate were incubated with 0.2 mL of physiological buffer containing 0.1 μCi/mL [$^{14}$C]DHA (incubation buffer) at 37˚C for 30 s to 15 min. At the end of the experiment, RBECs were washed with D-PBS three times and incubated with 0.2 mL of 1M NaOH (Wako; 192–02175) at 37˚C for 3 h for cell lysis. The total protein concentration in the cell lysates was determined using a Pierce™ BCA Protein Assay Kit. Samples were added to 10 mL of a liquid scintillation cocktail, then [$^{14}$C] DHA radioactivity in the cell lysate was measured using a liquid scintillation counter. The cellular uptake of [$^{14}$C]DHA by RBECs was expressed as cell/medium ratios calculated by dividing the radioactivity in one milligram of protein by the radioactivity in one microliter of the incubation buffer. The rate of cellular uptake was determined by the linear portion of the slope of the cell/medium ratios against the incubation time (0.5, 2, 5, and 10 min) graph.

## Western blot analysis

Equivalent amounts of protein from each sample were electrophoretically separated on 4–15% TGX Stain-Free gradient acrylamide gels (Bio-Rad, Hercules, CA; 4568084) or 12% TGX Stain-Free acrylamide gels (Bio-Rad; 161–0185) and transferred to low fluorescent polyvinylidene difluoride membranes (Bio-Rad; 1704274). Stain-Free technology using GelDoc go imaging system (Bio-Rad) was used for total protein normalization. Membranes were then blocked using Blocking One (Nacalai Tesque; 03953–95). MFSD2A, FABP5 and β-actin were detected using antibodies against MFSD2A (1:1,000; Sigma; SAB3500576), FABP5 (1:1,000; Sigma; SAB1401130) and β-actin (1:8,000; Sigma; A1978). After washing, the membranes were incubated in HRP–conjugated goat anti-rabbit IgG (Bio-Rad; 170–6515) or goat anti-mouse IgG (Bio-Rad; 170–6516), as appropriate. Immunoreactive bands were detected using Clarity Western ECL Substrate (Bio-Rad; 1705061). Images of the bands were digitally captured using a MultiImager II ChemiBOX (BioTools, Gunma, Japan), and band intensities were quantified using ImageJ software (National Institutes of Health Image, Bethesda, MD, USA). The relative

intensity of each individual protein was expressed as the ratio of the corresponding protein to the total protein loading or β-actin.

## Statistical analysis

Results are expressed as the mean ± standard error of the mean (SEM). Statistical analyses were performed using GraphPad Prism 8.0 (GraphPad, San Diego, CA, USA). Simple linear regression analysis was used to evaluate the brain uptake rate of [$^{14}$C]DHA and [$^{14}$C]sucrose in competition assay using excess unlabeled DHA and the rate of cellular uptake of [$^{14}$C]DHA. Statistical differences in brain uptake of [$^{14}$C]DHA and [$^{3}$H]mannitol and expression levels of MFSD2A and FABP5 protein were analyzed using one-way analysis of variance (ANOVA) followed by Tukey's multiple comparison tests. A two-way ANOVA (age × brain regions) was performed to analyze differences in the brain uptake of [$^{14}$C]DHA. An unpaired $t$-test was used to analyze the cellular uptake of [$^{14}$C]DHA and the expression levels of MFSD2A protein in RBECs transfected siMfsd2a. Differences were considered statistically significant for $P < 0.05$.

## Results

### Brain uptake of DHA decreased in 12- and 24-month-old mice

To evaluate the effects of aging on the brain uptake of DHA, [$^{14}$C]DHA brain/perfusate ratio was assessed in C57BL/6 mice at 2-, 8-, 12-, and 24 months of age. In the whole brains of mice aged 12 and 24 months, the brain uptake of [$^{14}$C]DHA decreased significantly compared with that of the 2-month-old group (Fig 1a). In mice at 12 and 24 months of age, the [$^{14}$C]DHA brain/perfusate ratio in the whole brain was decreased by 0.03275 mL/g ($P = 0.0284$) and 0.03827 mL/g ($P = 0.0380$), respectively. Given that aging decreased brain uptake of DHA, we assessed [$^{14}$C]DHA uptake by six brain regions (olfactory bulb, forebrain, cortex, hippocampus, thalamus and hypothalamus, and cerebellum) of 2-, 8-, 12-, and 24-month-old mice, to determine whether aging affects specific brain regions. The uptake of [$^{14}$C]DHA in the olfactory bulb, hippocampus, thalamus and hypothalamus of 12- and 24-month-old mice was considerably lower than that of 2-month-old mice. Two-way ANOVA revealed the significant effects of age (F(3,204) = 24.63, $P < 0.0001$) and regions (F(5,204) = 2.384, $P = 0.0396$), but there was no significant interaction between age and regions (F(15,204) = 0.5614, $P = 0.9016$).

One-way ANOVA showed an effect for age in each region as follows: whole brain (F = 4.422, $P = 0.0099$), olfactory bulb (F = 4.670, $P < 0.0077$), forebrain (F = 4.177, $P = 0.0127$), hippocampus (F = 9.209, $P = 0.0001$) and thalamus and hypothalamus (F = 6.458, $P = 0.0014$). The differences of [$^{14}$C]DHA brain/perfusate ratio in each region of the 12- and 24-month-old mice were as follows: 0.04626 mL/g (2 vs. 24 months old, $P = 0.0042$) in the olfactory bulb, 0.04489 mL/g (2 vs. 12 months old, $P = 0.0299$) and 0.05294 mL/g (2 vs. 24 months old, $P = 0.0375$) in the forebrain, 0.03402 mL/g (2 vs. 12 months old, $P = 0.0101$) and 0.06034 mL/g (2 vs. 24 months old, $P = 0.0001$) in the hippocampus and 0.04306 mL/g (2 vs. 12 months old, $P = 0.0051$), 0.05106 mL/g (2 vs. 24 months old, $P = 0.0067$) in the thalamus and hypothalamus (Fig 1b–1g).

### Brain uptake of mannitol did not decrease in 12- and 24-month-old mice

Next, we determined whether changes in BBB integrity during aging accounted for the decreased brain uptake of [$^{14}$C]DHA in mice aged 12- and 24- months. BBB integrity was evaluated using the brain/perfusate ratio of [$^{3}$H]mannitol at 1 min. We co-infused [$^{14}$C]DHA with [$^{3}$H]mannitol as a paracellular permeability marker as mannitol crosses the intact BBB poorly

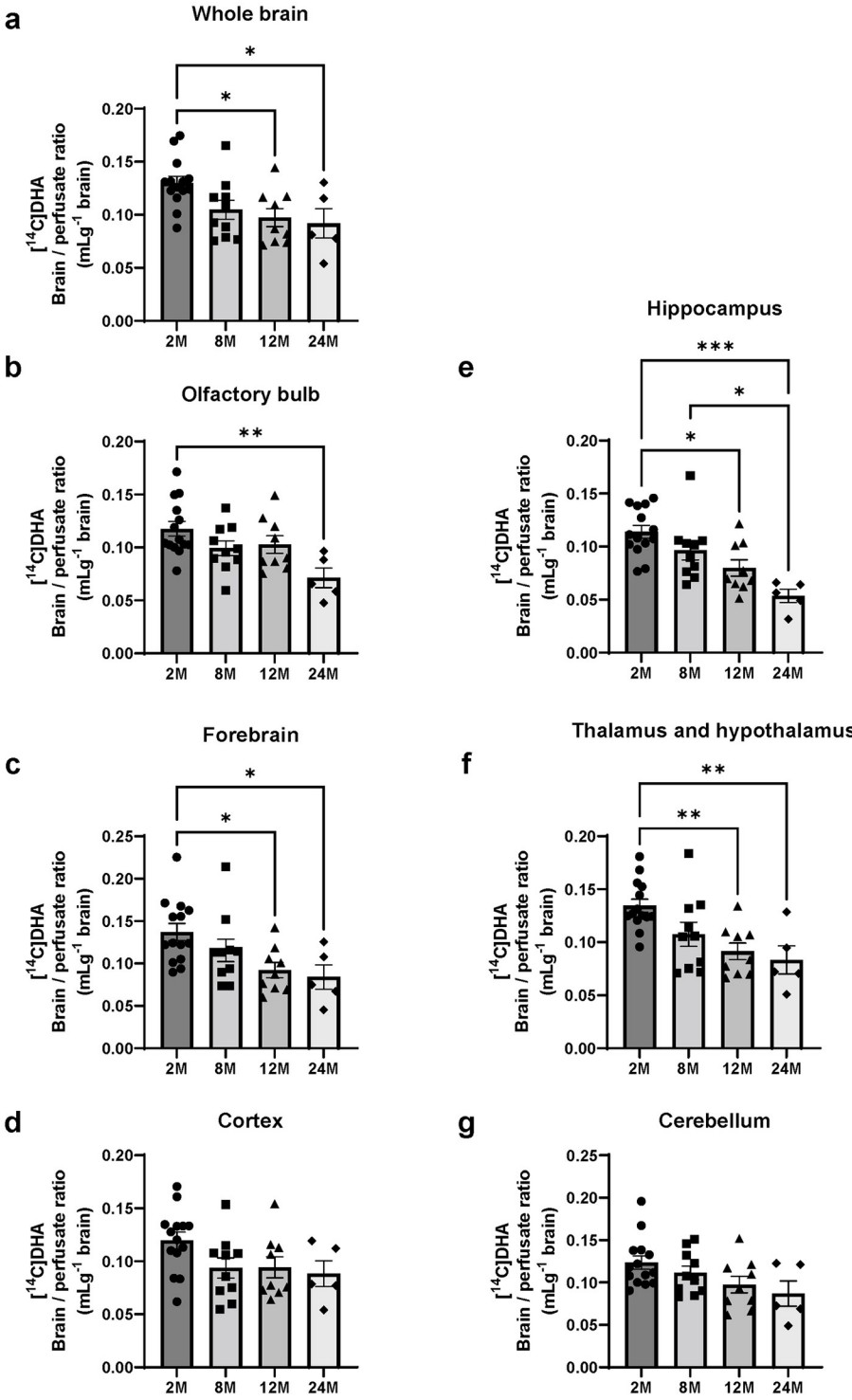

**Fig 1. Brain uptake of [$^{14}$C]docosahexaenoic acid (DHA) in 2-, 8-, 12-, and 24-month-old mice.** Brain/perfusate ratio of [$^{14}$C]DHA as NE-DHA in the whole brain (a), olfactory bulb (b), forebrain (c), cortex (d), hippocampus (e), thalamus and hypothalamus (f), and cerebellum (g) of 2- (2M), 8- (8M), 12- (12M), and 24-month-old (24M) mice following in situ transcardiac brain perfusion for 1 min at 2 mL/min. Data are shown as the mean ± standard error of the mean (n = 4–15). Each closed symbol represents an individual value. *P < 0.05, **P < 0.01, ***P < 0.001, significantly different from 2-month-old group.

[26]. No significant changes in brain uptake of [³H]mannitol were observed in the whole brain of 2-, 8-, 12-, and 24-month-old mice (Fig 2a). In each brain region, there are no significant effects with age (Fig 2b–2g).

## FABP5 protein expression in brain microvessels increased in 12- and 24-month-old mice

Following previous reports that brain NE-DHA uptake is mediated by passive membrane diffusion [12] and subsequent intracellular trafficking by an intracellular DHA carrier protein, FABP5 [14], we examined whether FABP5 expression is altered with aging. We evaluated the FABP5 protein expression levels in brain microvessels prepared from 2-, 8-, 12-, and 24-month-old mice using Western blot analysis. One-way ANOVA showed the effect of age on the FABP5 (F = 3.246, P = 0.0448) protein expression levels. The FABP5 protein expression in 24-month-old mice was significantly increased 4.6-fold (P = 0.0412) compared with that in the 2-month-old mice (Fig 3). These results indicate that the decreased brain uptake of [¹⁴C] DHA in aged mice was not caused by reduced FAPB5 expression. Since the previous study showed that the membrane permeability at the BBB did not change in 12- and 24-month-old mice [27], we ascertained whether NE-DHA is taken up into the brain by a transport carrier located on the luminal membrane of BMECs, such as MFSD2A.

## The brain uptake rate of DHA was decreased by adding excess unlabeled DHA

To determine the brain transport mechanism of NE-DHA, we measured the brain/perfusate ratio of [¹⁴C]DHA at 0.5–2.0 min using a perfusate containing unlabeled DHA 100 μM. Fig 4 shows a linear relationship between the brain/perfusate ratio of [¹⁴C]DHA and perfusion time up to 1.5 min in the whole brain and each brain region. The brain uptake of [¹⁴C]DHA plateaued at 1.5 min. The slope calculated by simple linear regression represents the brain uptake rate (Ki; mLg⁻¹min⁻¹) of [¹⁴C]DHA. For the whole brain, Ki of the DHA 100 μM group (Ki = 0.2921 mLg⁻¹min⁻¹) was significantly decreased compared with that of the vehicle group (Ki = 0.5944 mLg⁻¹min⁻¹, F = 10.89, P = 0.0036) (Fig 4a). In all brain regions, Ki of the DHA 100 μM group showed a significant reduction in comparison to the vehicle group (Fig 4b–4g). The data for all regions of the vehicle and DHA 100 μM groups are recorded below. Olfactory bulb (vehicle Ki: 0.6445 mLg⁻¹min⁻¹, DHA 100 μM Ki: 0.2848 mLg⁻¹min⁻¹, F = 6.959, P = 0.015), forebrain (vehicle Ki: 0.5712 mLg⁻¹min⁻¹, DHA 100 μM Ki: 0.2762 mLg⁻¹min⁻¹, F = 12.51, P = 0.021), cortex (vehicle Ki: 0.4516 mLg⁻¹min⁻¹, DHA 100 μM Ki: 0.2567 mLg⁻¹min⁻¹, F = 5.806, P = 0.0257), hippocampus (vehicle Ki: 0.5242 mLg⁻¹min⁻¹, DHA 100 μM Ki: 0.2478 mLg⁻¹min⁻¹, F = 7.369, P = 0.0133), thalamus and hypothalamus (vehicle Ki: 0.6684 mLg⁻¹min⁻¹, DHA 100 μM Ki: 0.2848 mLg⁻¹min⁻¹, F = 7.766, P = 0.0114), and cerebellum (vehicle Ki: 0.6837 mLg⁻¹min⁻¹, DHA 100 μM Ki: 0.3547 mLg⁻¹min⁻¹, F = 10.89, P = 0.0036).

## The BBB integrity was not altered by adding excess unlabeled DHA

Next, we determined whether changes in BBB integrity in the presence of excess unlabeled DHA accounted for the decreased brain uptake rate of [¹⁴C]DHA in the competition assay. BBB integrity was evaluated using the brain uptake rate of [¹⁴C]sucrose calculated by brain/perfusate ratio of [¹⁴C]sucrose at the corresponding perfusion time of the brain uptake of [¹⁴C] DHA, which revealed no differences between vehicle and DHA 100 μM groups (Fig 5a–5g). We paid attention to the concentration of the vehicle (ethanol) in the perfusate since a cerebral vascular volume (the y-intercept of a regression line of brain/perfusate ratio of [¹⁴C]sucrose

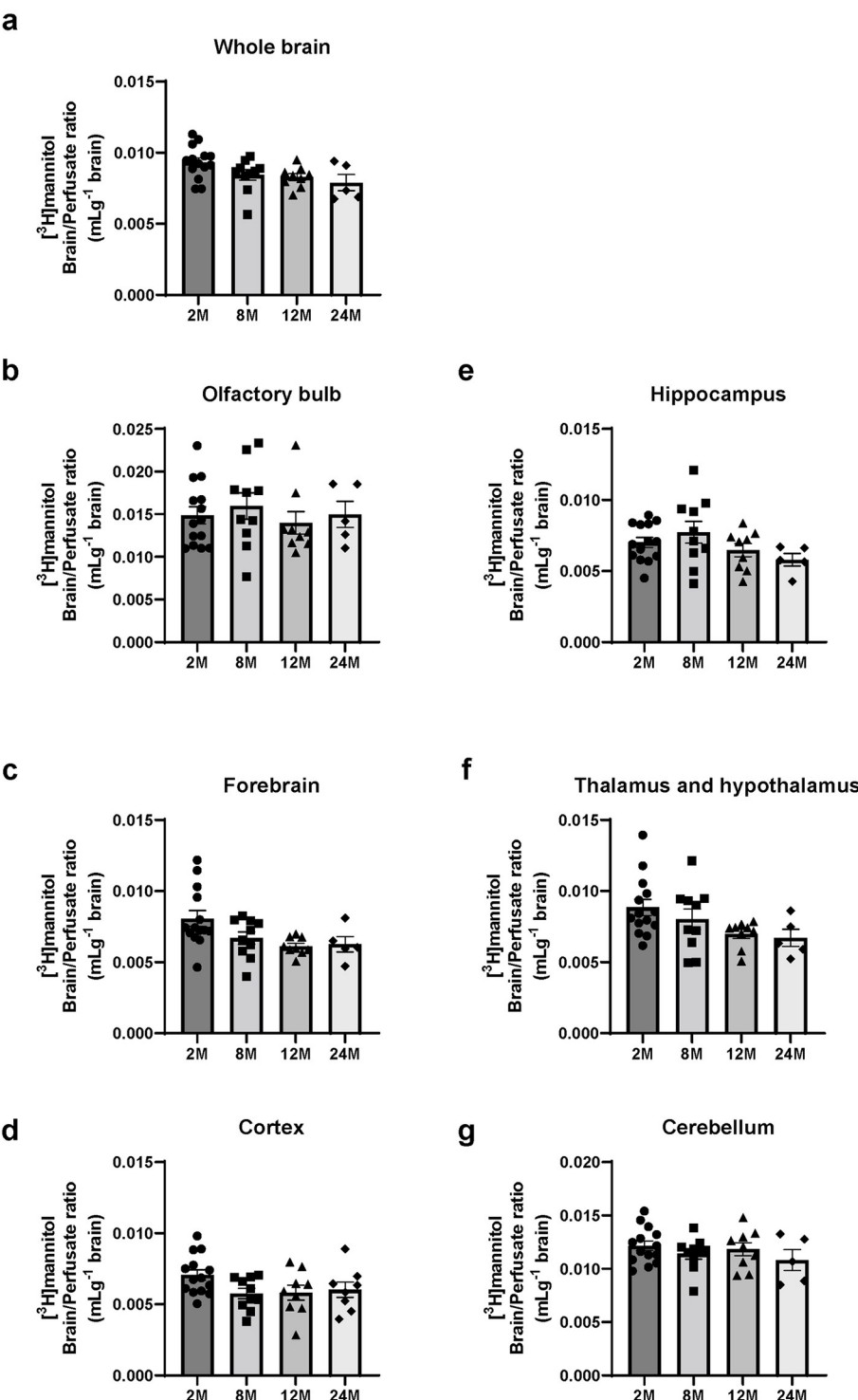

**Fig 2. Brain uptake of [³H]mannitol in 2-, 8-, 12-, and 24-month-old mice.** Brain/perfusate ratio of [³H]mannitol in the whole brain (a), olfactory bulb (b), forebrain (c), cortex (d), hippocampus (e), thalamus and hypothalamus (f), and cerebellum (g), of 2- (2M), 8- (8M), 12- (12M), and 24-month-old (24M) mice following in situ transcardiac brain perfusion for 1 min at 2 mL/min. Data are shown as the mean ± standard error of the mean (n = 4–15). Each closed symbol represents an individual value.

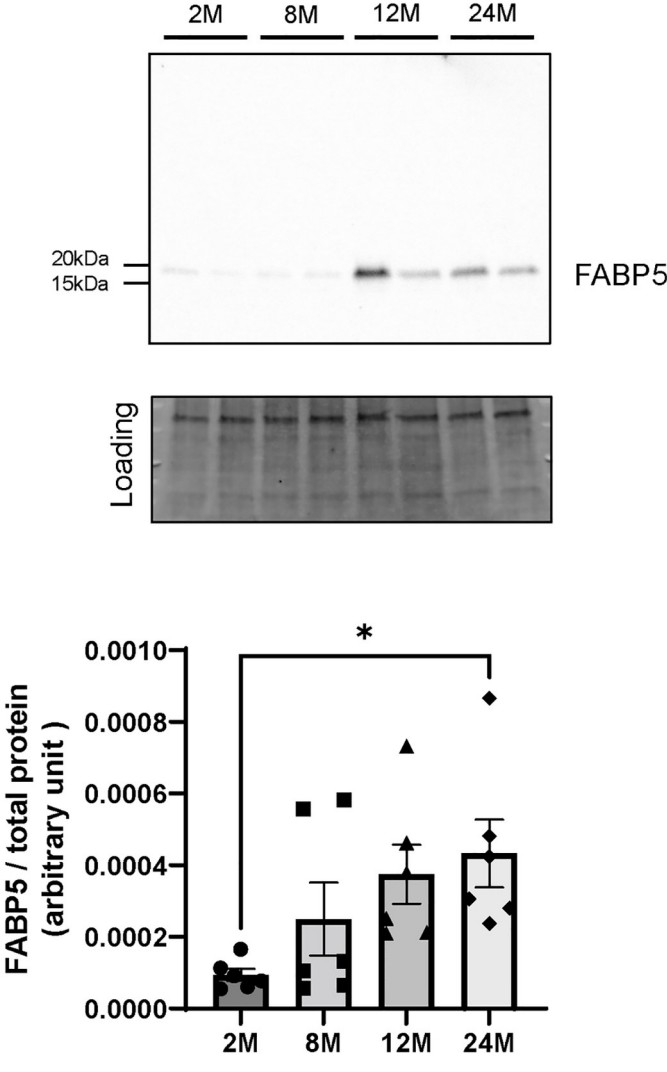

**Fig 3. FABP5 expression levels in brain microvessels from 2-, 8-, 12-, and 24-month-old mice.** Representative Western blot images and band intensities quantified by densitometry in 2- (2M), 8- (8M), 12- (12M), and 24-month-old (24M) mice. Total protein levels measured by Stain-free technology were used as the loading controls for total protein normalization. Bars indicate the mean ± standard error of the mean (n = 5–6). Each closed symbol represents an individual value. *P < 0.05, significantly different from the 2-month-old group. Abbreviations: FABP5, fatty acid-binding protein 5.

over perfusion time) enlarged by a high concentration of ethanol (> 0.001%) resulted in the uncertainty of the inhibition effect of unlabeled DHA.

## MFSD2A mediated the cellular uptake of DHA by brain endothelial cells

To determine whether transport carriers mediate the brain uptake of NE-DHA and if MFSD2A is involved in the brain uptake of NE-DHA, we examined the effects of temperature and siMfsd2a on cellular uptake of [$^{14}$C]DHA by BMECs. Fig 6a illustrates the time course of cellular uptake of [$^{14}$C]DHA by RBECs. Because cellular uptake of [$^{14}$C]DHA by RBECs peaked at 10 min, the following experiments were performed at 10 min. The slope calculated by simple linear regression represents the rate of cellular uptake (μLmg protein$^{-1}$ min$^{-1}$) of [$^{14}$C]DHA. The rate of cellular uptake of [$^{14}$C]DHA by RBECs at 4˚C was significantly

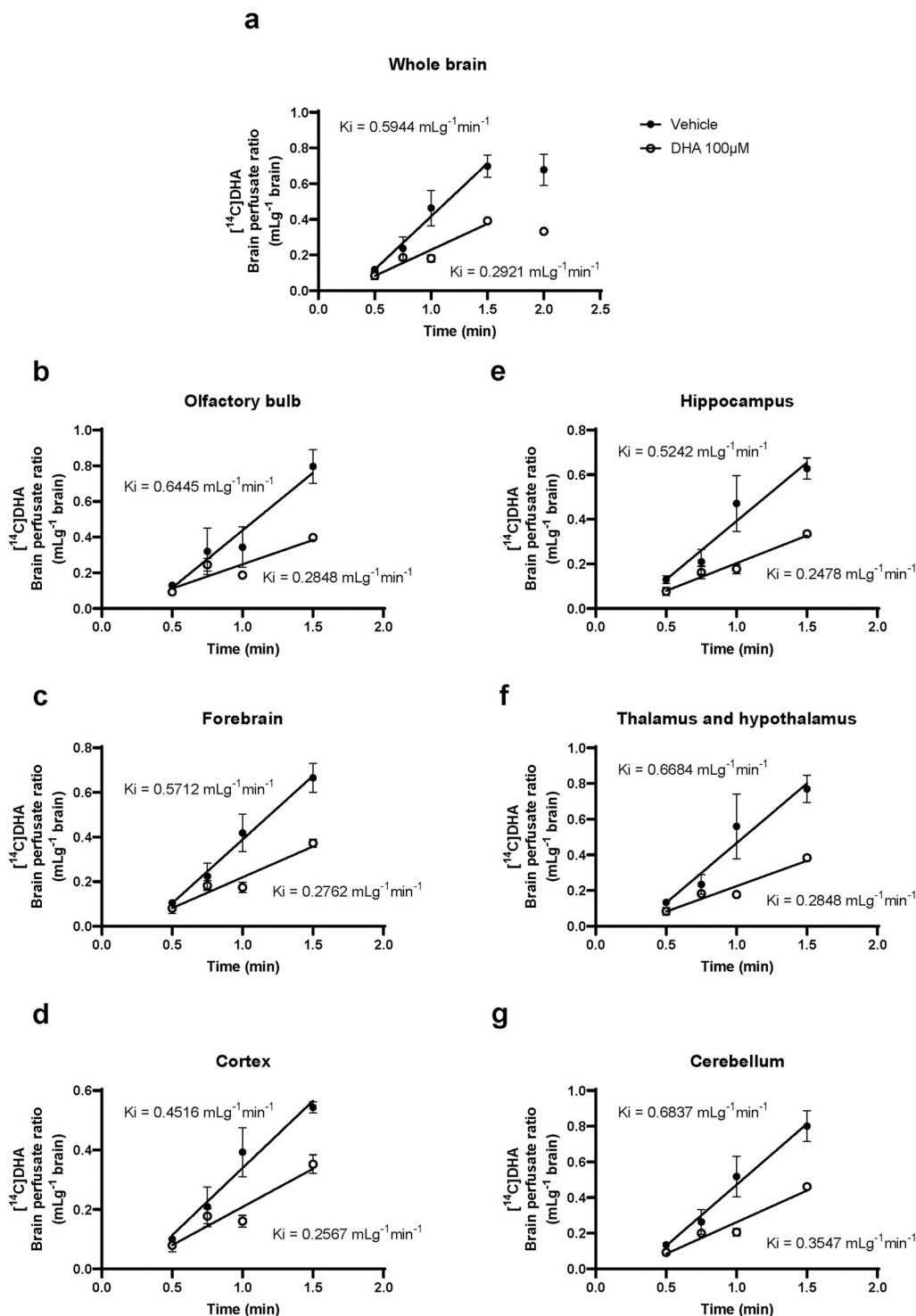

**Fig 4. Brain uptake rate of [14C]docosahexaenoic acid (DHA) in the whole brain and each brain region.** Brain uptake of [14C]DHA is expressed as brain/perfusate ratios following *in situ* transcardiac brain perfusion at 2 mL/min in 2-month-old mice. Linear regression of the mean brain/perfusate ratio of [14C]DHA (mL/g brain) over perfusion time (0.5, 0.75, 1, and 1.5 min) in whole brain (a), olfactory bulb (b), forebrain (c), cortex (d), hippocampus (e), thalamus and hypothalamus (f), and cerebellum (g) of vehicle and DHA 100 μM groups yielded Ki (mL/g brain • $min^{-1}$). Ki represents the brain uptake rate of [14C]DHA. Data are shown as the mean ± standard error of the mean (n = 3/time point).

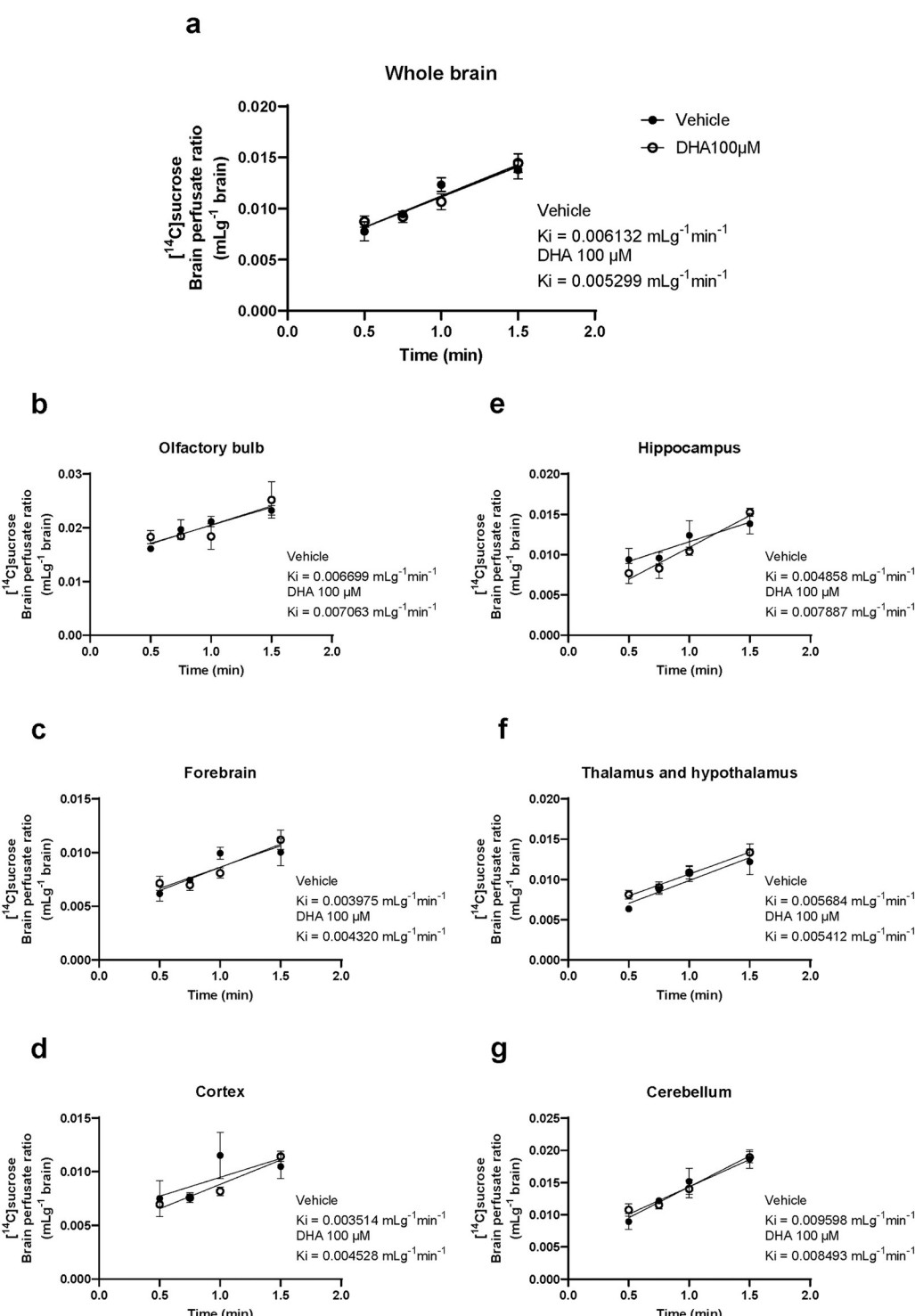

**Fig 5. Brain uptake rate of [14C]sucrose in the whole brain and individual brain regions.** Brain uptake of [14C]sucrose is expressed as brain/perfusate ratios following *in situ* transcardiac perfusion at 2 mL/min in 2-month-old mice. Linear regression of the mean brain/perfusate ratio of [14C]sucrose over perfusion time (0.5, 0.75, 1, and 1.5 min) in whole brain (a), olfactory bulb (b), forebrain (c), cortex (d), hippocampus (e), thalamus and hypothalamus (f), and cerebellum (g) of vehicle and DHA 100 μM groups yielded Ki (mL/g brain • min⁻¹). Ki represents the brain uptake rate of [14C]sucrose. Data are shown as the mean ± standard error of the mean (n = 3/time point).

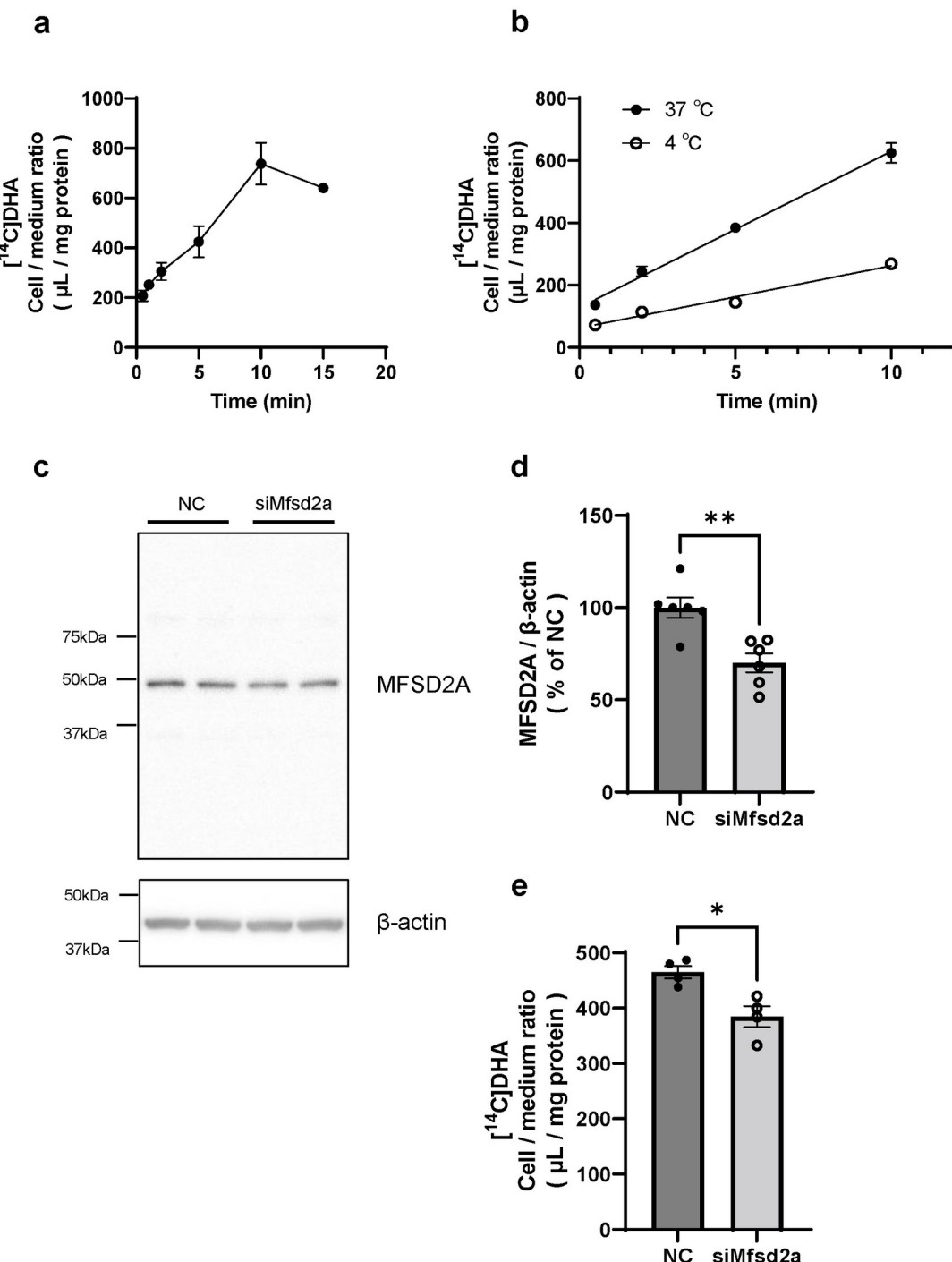

**Fig 6. Cellular uptake of [¹⁴C]docosahexaenoic acid (DHA) by RBECs transfected with MFSD2A siRNA.** (a)Time course of cellular uptake of [¹⁴C]DHA (μL/mg protein) by intact RBECs (n = 4–6). (b) Time course of cellular uptake of [¹⁴C]DHA (μL/mg protein) by intact RBECs at 37˚C and 4˚C (n = 3). (c, d) Effect of Mfsd2a siRNA transfection on MFSD2A protein expression in RBECs. RBECs were treated with transfection reagent and 50 nM Mfsd2a siRNA for 48 h. Panel (c) shows representative Western blot images. Panel (d) shows the quantified band intensities corrected by β-Actin as the loading control in RBECs transfected with negative control siRNA (NC) and Mfsd2a siRNA (siMfsd2a). (n = 6) (e) Cellular uptake of [¹⁴C]DHA by RBECs transfected with negative control (NC) and Mfsd2a siRNA (n = 4). The cellular uptake of [¹⁴C]DHA for 2 min is expressed as cell/medium ratio (μL/mg protein). Data are shown as the mean ± standard error of the mean. Each closed symbol represents an individual value. *P < 0.05, **P < 0.01, significantly different from negative control siRNA-transfected RBECs. Abbreviation: RBECs, rat brain endothelial cells. MFSD2A, major facilitator superfamily domain-containing protein 2A.

decreased by 60% (19.98 μLmg protein$^{-1}$ min$^{-1}$) compared with the uptake rate at 37°C (50.13 μLmg protein$^{-1}$ min$^{-1}$, F = 112.8, P < 0.0001) (Fig 6b). The MFSD2A expression level in RBECs transfected with siMfsd2a was significantly down-regulated by 30% (P = 0.0026) compared with that of the cells transfected with negative control siRNA (Fig 6c and 6d). The siMfsd2a-transfected RBECs showed a significant decrease of 17% (P = 0.0108) in cellular uptake of [$^{14}$C]DHA compared with negative control siRNA-transfected cells (Fig 6e).

## MFSD2A protein expression in brain microvessels decreased in 12- and 24-month-old mice

We determined the MFSD2A protein expression levels in brain microvessels prepared from 2-, 8-, 12-, and 24-month-old mice using Western blot analysis. A previous study showed that multiple MFSD2A immunoreactive bands of approximately 55 kDa are detected in lysates from mouse brains and neural stem cells [28]. Therefore, we considered all detected bands with an approximate size of 50 kDa as immunoreactive MFSD2A bands and quantified their intensity. One-way ANOVA showed the effect of age in the MFSD2A (F = 5.171, P = 0.0109) protein expression levels. The 12- and 24-month-old mice exhibited significant reductions in MFSD2A protein expression of 29% (12-month-old mice, P = 0.0242) and 26% (24-month-old mice, P = 0.0432), respectively, compared with 2-month-old mice (Fig 7).

## Discussion

An age-related decline in brain DHA is associated with cognitive decline [18, 20]. Because brain DHA levels in adulthood are largely dependent on direct dietary intake, DHA in the peripheral circulation needs to be efficiently transported into the brain across the BBB [7]. DHA is present in plasma in a bound form; therefore, the cellular uptake of DHA depends on its affinity for plasma proteins [29]. In the present study, to exclude the influence of plasma protein binding to DHA, we used a transcardiac brain perfusion technique that is commonly used to evaluate BBB permeability [30]. The concentration of [$^{14}$C]DHA in the perfusate was the same as that reported in a previous study [31].

Using the perfusion technique, we demonstrated a significant decrease in the BBB transport of [$^{14}$C]DHA in the whole brains of 12- and 24-month-old mice compared with that in 2-month-old mice (Fig 1). To evaluate region specificity in the brain uptake of [$^{14}$C]DHA, the brain was divided into six regions. There were age-related decreases in the brain uptake of [$^{14}$C]DHA in specific regions (Fig 1). These data suggest that the availability of DHA is not influenced by aging in the cortex and cerebellum.

Furthermore, we evaluated the brain/perfusate ratio of [$^{3}$H]mannitol, a small hydrophilic molecule with low paracellular permeability across the BBB. The brain/perfusate ratio of [$^{3}$H]mannitol reflects the cerebral vascular volume, because [$^{3}$H]mannitol does not cross the BBB during a short period and remains within the intracerebral vasculature. There were no significant differences in brain uptake of [$^{3}$H]mannitol among the 2-, 8-, 12-, and 24-month-old groups. Our data is supported by a previous study that reported that BBB permeability did not alter in 28-month-old aged rats [32]. However, previous works showed the leakiness of human BBB in normal aging [33, 34]. These discordances in age-related changes in BBB permeability probably depend on the evaluation methods used, including markers used for BBB permeability. Our findings suggest that the decreased brain uptake of [$^{14}$C]DHA in 12- and 24-month-old mice is not the result of changes in BBB integrity or cerebral vascular volume. These results indicate that the decreased DHA transport across the BBB in aged mice is predominantly attributable to changes in a transcellular pathway of DHA involving carrier-mediated transport and passive diffusion rather than a paracellular pathway.

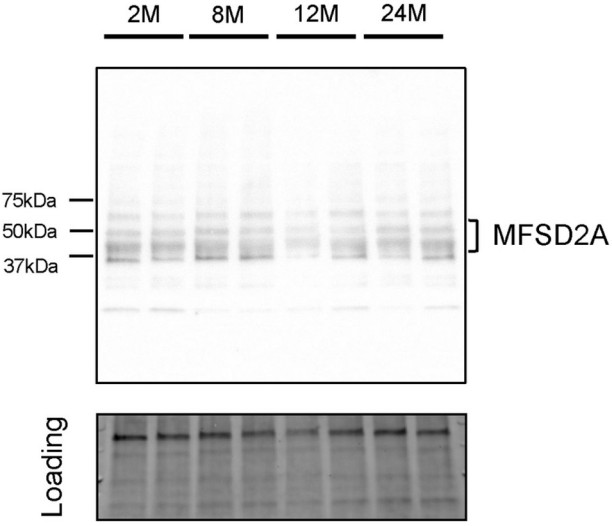

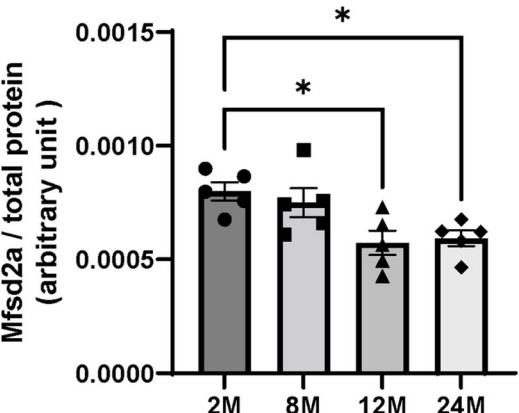

**Fig 7. MFSD2A expression levels in brain microvessels from 2-, 8-, 12-, and 24-month-old mice.** Representative Western blot images and band intensities quantified by densitometry in 2- (2M), 8- (8M), 12- (12M), and 24-month-old (24M) mice. Total protein levels measured by Stain-free technology were used as the loading controls for total protein normalization. Bars indicate the mean ± standard error of the mean (n = 5–6). Each closed symbol represents an individual value. $^*P < 0.05$, significantly different from 2-month-old group.

Given that DHA crosses BMECs via a transcellular pathway, a possible explanation for the age-related decreased brain uptake of DHA is the decline in the FABP5-mediated intracellular trafficking of DHA. Interestingly, we observed that FABP5 protein expression in the microvessel increased with age (Fig 3). The highest level of FABP5 expression is detected during neonatal development and then declines after birth [35], indicating that FABP expression in the brain is altered with age [36]. Although further studies are needed to clarify whether the intracellular DHA transport activity of FABP5 is maintained during aging, our data suggest that the age-related decreased brain uptake of NE-DHA is likely due to compromised influx of NE-DHA through the luminal membrane of BMECs rather than the decline in the intracellular NE-DHA trafficking by FABP5. Therefore, we ascertained that NE-DHA is taken up into the brain by a transport carrier in the following experiments.

We demonstrated that the brain uptake of [14C]DHA was inhibited by excess unlabeled NE-DHA (Fig 4) without changes in the brain/perfusate ratios of [14C]sucrose (MW: 342.3), a small hydrophilic molecule with low paracellular permeability across the BBB, as a marker of cerebral vascular volume and BBB integrity, in all the brain regions (Fig 5). Therefore, [14C]DHA transport across the BBB is mediated by a saturable transport system. Our data are partly consistent with the findings of a previous study indicating that [14C]DHA is transported across the BBB by passive diffusion [12], as the extent of inhibition by excess (55-fold) unlabeled DHA accounted for ~50% of the brain uptake rate of [14C]DHA.

Next, we determined whether transport carriers mediate the brain uptake of [14C]DHA and if MFSD2A is involved in the brain uptake of [14C]DHA at the BBB using RBECs. We confirmed that RBECs expressed MFSD2A (Fig 6c). In cellular uptake experiments, we used a physiological buffer containing [14C]DHA without any proteins or phospholipids, indicating that [14C]DHA was taken up by RBECs as the non-esterified form. We found that the cellular uptake of [14C]DHA by RBECs was temperature-dependent (Fig 6b). This result indicated that [14C]DHA is taken up by transport carriers on BMECs and supported our data using the transcardiac brain perfusion technique (Fig 4). MFSD2A knockdown by siRNA decreased MFSD2A protein levels and cellular uptake of [14C]DHA by RBECs (Fig 6c–6e). Taking into account previous studies showing that LPC-DHA is a primary substrate for MFSD2A [16] and FABP5 contributes to the intracellular transport of NE-DHA, which penetrates the luminal membrane of BMECs [14], our results suggest that MFSD2A may also serve as a transporter for extracellular NE-DHA.

Considering our in vitro results that siRNA-mediated knockdown of MFSD2A decreased brain endothelial uptake of [14C]DHA, another possible explanation for the decrease in DHA uptake in the aged brain is that MFSD2A protein expression on the luminal surface of the microvasculature is down-regulated. MFSD2A is expressed in CNS vasculature in the entire brain [37]. In line with this concept, our results showed that the microvascular expression of MFSD2A is down-regulated in the 12- and 24-month-old brains (Fig 7) which is consistent with the reduced [14C]DHA uptake by the whole brain (Fig 1a). In addition, we measured mRNA expression levels of MFSD2A in brain microvessels from 24-month-old mice and confirmed that they were down-regulated compared with those from 2-month-old mice (S1 Fig). Our data is supported by a previous study demonstrating that MFSD2A protein expression levels of the brain endothelial cells decreased in aged mice [38]. Therefore, it is possible that FABP5 up-regulation in 12- and 24-month-old mice (Fig 3) may play a crucial role in aged mice for maintaining brain DHA levels and regulating brain uptake of DHA. Further studies are required to clarify the mechanisms underlying age-related changes in MFSD2A and FABP5 expression. A limitation of our study is that we were forced to measure the expression levels of MFSD2A in microvessels obtained from the whole brain, but not the brain regions of interest due to a low yield of isolated microvessels from each brain region. We observed the regional variations of brain uptake of [14C]DHA with aging (Fig 1b–1g). Therefore, the age-related decrease in the expression levels of MFSD2A in the olfactory bulb, hippocampus, and thalamus and hypothalamus might be greater than in the other regions. In addition, we cannot exclude the possibility that localization patterns of MFSD2A in BMECs are altered with aging, leading to impaired NE-DHA transport activity of MFSD2A. Future studies may shed light on the regional variations of the age-related down-regulation and/or impaired functional activity of MFSD2A.

There are technical limitations in this study. First, a transcardiac brain perfusion technique was optimized for fatty acids by Pan et al. [14]. Since the perfusion rate was given at 2 mL/min, without adding BSA in the perfusate, the cerebral vascular pressure was insufficient. The perfusion rate of 10 mL/min is more appropriate. The brain/perfusate ratio of [14C]DHA in the present study is lower than that reported by Pan et al. [14]. Therefore, our results of the Kin value and/or brain/perfusate ratio of [14C]DHA may be underestimated or not accurate due to

insufficient cerebral vascular pressure caused by a lower perfusion rate. Of note, we found that the aged brain exhibited a decreased uptake of [14C]DHA compared with the young brain even if the flow rate of 2 mL/min did not give sufficient cerebral vascular pressure. Therefore, further studies are needed to determine whether a higher perfusion rate (10 mL/min) would affect the obtained results. Second, in this study, we could not evaluate how the [14C]DHA was processed. Therefore, it is unclear whether [14C]DHA was transported across the BBB in the non-esterified form. Previous works using HPLC analysis showed that the majority of radioactivity was detected in total phospholipid fractions of [14C]DHA-perfused brains corresponding to DHA in the perfusate and any radiolabeled compounds associated with DHA were not detected after a 40 s brain perfusion [12, 39]. In addition, capillary depletion of brain homogenates after brain perfusion showed that less than 10% of [14C]DHA remained in endothelial cells of the brain vasculature [12]. Therefore, we considered that [14C]DHA was probably transported across the BBB as NE-DHA. However, we cannot exclude the possibility that [14C] DHA derivatives are also quantified.

Several studies have demonstrated the relationship between brain DHA levels and events that occur with increased age. For example, patients with dementia have decreased brain DHA [30, 40], and DHA supplementation improves the accumulation of brain DHA in a dementia mouse model [41, 42]. Further, higher DHA intake is inversely correlated with the relative risk of AD [43]. In addition, a decrease in DHA levels likely contributes to the cognitive impairment that is observed in individuals with AD [18]. MFSD2A expression levels in brain endothelial cells in AD patients were lower than those in healthy older adults [38], suggesting that insufficient transport of DHA across the BBB is one of the contributing factors underlying lower brain DHA levels in AD [44]. Indeed, the BBB transport of DHA is decreased in a mouse model for AD [45]. Notably, although 10–12 month-old mice do not exhibit cognitive impairment [46], we demonstrated a decrease in DHA transport across the BBB in the hippocampus at this age in the present study. Together, these findings suggest that deficient DHA transport across the BBB precedes age-related cognitive decline. Several clinical studies have indicated that DHA supplementation has benefits for cognitive health in aging [47], but the required oral dose of DHA supplementation for brain delivery remains unknown. The possibility that aging lowers the transport activity of DHA at the BBB should be considered when DHA supplementation is offered to older adults. An effective intervention that delivers DHA to the brain may be successful in improving brain DHA bioavailability in older adults.

## Conclusions

In conclusion, we report reduced brain uptake of [14C]DHA in middle-aged (12-month-old) and aged (24-month-old) mice. Furthermore, we demonstrated that [14C]DHA is transported across the BBB by a saturable transport system and that MFSD2A partly mediates brain endothelial uptake of [14C]DHA. Finally, we observed a decreased expression of MFSD2A in microvessels obtained from middle-aged and aged mice. Therefore, these findings suggest that the reduced brain uptake of DHA in middle-aged and aged mice could be attributable to age-related down-regulation of MFSD2A, but not FABP5. These results suggest that improving deficient DHA transport across the BBB in older adults by DHA supplementation could be a new approach to enhance the therapeutic efficiency of treatment for AD or age-related cognitive decline.

## Supporting information

**S1 Fig. The mRNA expression levels of Mfsd2a in brain microvessels from 2- and 24-month-old mice.** The mRNA expression levels for Mfsd2a in brain microvessels from 2-

(2M) and 24-month-old (24M) mice were quantified by real-time quantitative PCR. Data are shown as the mean ± standard error of the mean. Each closed symbol represents an individual value (n = 9–10). **P < 0.01, significantly different from 2-month-old group.
(TIFF)

**S1 File. Supplementary method.**
(DOCX)

**S1 Raw images.**
(PDF)

## Acknowledgments

We would like to thank Editage (www.editage.com) for English language editing.

## Author Contributions

**Conceptualization:** Takuro Iwao, Shinya Dohgu.

**Data curation:** Takuro Iwao, Shinya Dohgu.

**Formal analysis:** Takuro Iwao, Shinya Dohgu.

**Funding acquisition:** Takuro Iwao, Fuyuko Takata, Shinya Dohgu.

**Investigation:** Takuro Iwao, Fuyuko Takata, Junichi Matsumoto, Hisataka Aridome, Miho Yasunaga, Miki Yokoya, Shinya Dohgu.

**Methodology:** Takuro Iwao, Shinya Dohgu.

**Supervision:** Yasufumi Kataoka, Shinya Dohgu.

**Validation:** Takuro Iwao, Fuyuko Takata, Shinya Dohgu.

**Visualization:** Takuro Iwao, Junichi Matsumoto, Shinya Dohgu.

**Writing – original draft:** Takuro Iwao.

**Writing – review & editing:** Fuyuko Takata, Junichi Matsumoto, Yasufumi Kataoka, Shinya Dohgu.

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
