## [Decision Letter · Decision Letter 0]

29 Nov 2022

PONE-D-22-31018Aging decreases docosahexaenoic acid transport across blood-brain barrier in C57BL/6J micePLOS ONE

Dear Dr. Dohgu,

Thank you for submitting your manuscript to PLOS ONE. After careful consideration, we feel that it has merit but does not fully meet PLOS ONE’s publication criteria as it currently stands. Therefore, we invite you to submit a revised version of the manuscript that addresses all the points raised during the review process.

Two experts evaluated the manuscript. Both of them found the paper interesting and valuable and suggested amendments. No new experiments are needed, but new calculations/normalizations may be needed,  the limitations of the methods and experiments should be addressed in details (Discussion) and the conclusions should be drawn more carefully.

We look forward to receiving your revised manuscript.

Kind regards,

Mária A. Deli, M.D., Ph.D.

Academic Editor

PLOS ONE

Journal Requirements:

Reviewers' comments:

Reviewer's Responses to Questions

**Comments to the Author**

1. Is the manuscript technically sound, and do the data support the conclusions?

Reviewer #1: Partly

Reviewer #2: Partly

2. Has the statistical analysis been performed appropriately and rigorously? 

Reviewer #1: Yes

Reviewer #2: Yes

3. Have the authors made all data underlying the findings in their manuscript fully available?

Reviewer #1: Yes

Reviewer #2: Yes

4. Is the manuscript presented in an intelligible fashion and written in standard English?

Reviewer #1: Yes

Reviewer #2: Yes

5. Review Comments to the Author

Reviewer #1: The paper is technically sound and interesting. The main concern is that no appropriate analytical method was used to prove that DHA is indeed in its non-esterified structure. Once living tissue is involved, either cells or in-vivo you cannot be certain that the DHA is not metabolized rapidly by secreted/cellular factors. Although the media in the cells experiments was devoid of proteins and PL and in-vivo the authors used the transcardiac brain perfusion technique, these can only increase the possibility of DHA not being esterified before the transport to/by the EC. This is of course even more important when looking at DHA levels inside the brain after it has been transported via the BBB; using radiolabeled DHA and measuring it in brain tissue doesn’t guarantee that what you “see” is NE-DHA. You can only conclude that the CPM levels were increased and assume that it is related to some form of DHA. (In which position was the DHA labelled exactly with C14?). So, the author conclusion is not based on experimental evidence rather on assumptions. Basically, the use of [14C] alone cannot inform about the physiological process of DHA uptake by the transporter:

1- Not knowing in which C position is the label, it is impossible to understand how the molecule was processed: as a structural component or as a beta oxidation substrate?

2-The labelled DHA may have gone through re-esterification in the system (in vivo or in vitro) and therefore it is no longer unesterified.

Besides that, the experiments are sound and the data about mfsd2a and fabp5 levels in aging mice is interesting and important. But again, also here, the conclusions are too strong and need to be tuned down. For example, the increase in fabp4 (4 fold) is very impressive suggesting that this protein might be important in the aged brain. Is it possible that its activation is reduced with age therefore the cells increase its production? the authors need to work on a better model which is more related to the molecular events they intend to describe. In the present form, the conclusions are based on inferences rather than facts.

Other issues to address:

1. Please give evidence (or at least reference) for the purification level of the isolated brain blood vessels (IBV). How can you be sure that the results obtained with IBV are not derived from contaminating non-EC cells in the BV fraction? To be more certain about the conclusions derived from experiments with BV, one should conduct parallel experiments with highly purified BEC for example.

2. How do you account for the lack of decrease in BBB permeability to mannitol with age? There is plenty of literature regarding leakiness of the BBB in normal aging, especially in the Hipp (see Zlokovic papers for example). This point needs to be better discussed.

3. Please show the plateau obtained in the uptake experiments (line 395)

4. Lines: 67-71: This statement regarding ALA is controversial. Whereas dietary maritime DHA supply has been regarded as the main efficient source of brain DHA the ability of its metabolic precursor, terrestrial ALA, to support brain DHA (as well as EPA), has rarely been examined. Recent studies have begun to indicate that ALA enrichment in adult diets and in maternal diets during fetal development and weaning increases brain DHA levels and higher expression of Mfsd2a indicating ALA ability to enable higher DHA levels and improved BBB transport. Please refer to this literature.

5. Why is sucrose used instead of mannitol? Please discuss the differences between the 2 markers and state the reason for using each marker in the different experiments

6. In the cellular uptake experiments after transfection, do you account for cellular death? Any other factors that may affect uptake which are not related to mfsd2a levels? Did you try pharmacological inhibition of the transporter?

7. References missing in line 513 regarding DHA and cognitive decline

8. When counting C14 inside the brain parenchyma, how do you make sure the C14 doesn’t originate from the BV in this tissue without BV depletion? Please describe in more details the brain perfusion technique and why do you think it provides the experimental conditions for the NE-DHA to remain NE. Is it possible that the injected DHA will be metabolized before binding to mfsd2a on the BEC? The same for the in-vitro studies. See PMID: 9886086, as an example: it is a useful reference to understand how a follow up of DHA location may be done.

9. Lines 527-529 in discussion: did you test for significance between the regions? Please do, otherwise you cant say that.

10. The term “vascular space” should be better defined

11. Lines 581-582: this conclusion doesn’t stem from your results. You didn’t check it.

12. To strengthen the conclusions derived from the in-vitro studies one should add at least one of the following: competition with cold DHA, pharmacological inhibition, use other substrates which are ligands for other transporters and show they are not affected by the transfection. And again, measurements of radioactivity and not directly DHA using other analytical methods (HPLC and GC) can only provide assumptions and not proofs of what is the exact DHA form in the different stages/regions.

Reviewer #2: Major:

The in situ transcardiac perfusion was developed by William Banks laboratory, and later optimized for fatty acid by Pan et al. (https://pubs.acs.org/doi/10.1021/acs.molpharmaceut.5b00580). Please also include Pan et al as reference here (i.e. current reference 13). Can the authors confirm if they have included BSA in the perfusion fluid, as otherwise 2 mL/min may not give sufficient pressure for proper perfusion. If BSA was not included, please justify if 2 mL/min is appropriate.

Fig 1 & 2, please express B:P ratio as mL/mg OR mL/g, rather than normalizing all values to that of 2 months old as a %.

Fig 3 - The MW of FABP5 is ~ 15 kDa, but it was labeled as 25 kDa. Is this a mistake? Again, do not normalize to 2 month old.

Fig 4 - Please change the unit to mL/g rather than uL/g

I am unsure if it will change the conclusion without normalizing to 2 month old. I will comment on the discussion when I see the revised manuscript.

Minor

Line 119-120, please specify isotope for DHA, sucrose in the subheading

6. PLOS authors have the option to publish the peer review history of their article (what does this mean?). If published, this will include your full peer review and any attached files.

Reviewer #1: No

Reviewer #2: **Yes: **Dr Yijun Pan

---

## [Author Response · Author response to Decision Letter 0]

16 Jan 2023

Response to Reviewers

Reviewer #1: 

We appreciate the reviewer’s comments and suggestions, which have enabled us to considerably improve our manuscript. As indicated in the following responses, we have considered all of these comments and suggestions in the revised version of our manuscript.

The paper is technically sound and interesting. The main concern is that no appropriate analytical method was used to prove that DHA is indeed in its non-esterified structure. Once living tissue is involved, either cells or in-vivo you cannot be certain that the DHA is not metabolized rapidly by secreted/cellular factors. Although the media in the cells experiments was devoid of proteins and PL and in-vivo the authors used the transcardiac brain perfusion technique, these can only increase the possibility of DHA not being esterified before the transport to/by the EC. This is of course even more important when looking at DHA levels inside the brain after it has been transported via the BBB; using radiolabeled DHA and measuring it in brain tissue doesn’t guarantee that what you “see” is NE-DHA. You can only conclude that the CPM levels were increased and assume that it is related to some form of DHA. (In which position was the DHA labelled exactly with C14?). So, the author conclusion is not based on experimental evidence rather on assumptions. Basically, the use of [14C] alone cannot inform about the physiological process of DHA uptake by the transporter:

1- Not knowing in which C position is the label, it is impossible to understand how the molecule was processed: as a structural component or as a beta oxidation substrate?

2-The labelled DHA may have gone through re-esterification in the system (in vivo or in vitro) and therefore it is no longer unesterified.

Besides that, the experiments are sound and the data about mfsd2a and fabp5 levels in aging mice is interesting and important. But again, also here, the conclusions are too strong and need to be tuned down. For example, the increase in fabp4 (4 fold) is very impressive suggesting that this protein might be important in the aged brain. Is it possible that its activation is reduced with age therefore the cells increase its production? the authors need to work on a better model which is more related to the molecular events they intend to describe. In the present form, the conclusions are based on inferences rather than facts.

Response: We thank the reviewer for the comment. As the reviewer mentioned, we used the radiolabeled [1-14C]DHA, but did not evaluate how the labeled DHA was processed in this study. Previous works using HPLC analysis showed that the majority of radioactivity was detected in total phospholipid fractions of [14C]DHA-perfused brains corresponded to DHA in perfusate and any radiolabeled compounds associated with DHA were not detected after perfusion, whereas [14C]EPA was indeed rapidly β-oxidized (PMID: 19237271, 19442696). Therefore, we considered that [14C]DHA was probably transported across BBB as NE-DHA. According to the reviewer’s suggestion, the conclusion was turned down and the limitation regarding [14C]DHA transport was added as follows: 

Page 36, Lines 618-628: …. the obtained results. Second, in this study, we could not evaluate how the [14C]DHA was processed. Therefore, it is unclear whether [14C]DHA was transported across the BBB in the non-esterified form. Previous works using HPLC analysis showed that the majority of radioactivity was detected in total phospholipid fractions of [14C]DHA-perfused brains corresponding to DHA in the perfusate and any radiolabeled compounds associated with DHA were not detected after a 40 s brain perfusion [39,40]. In addition, capillary depletion of brain homogenates after brain perfusion showed that less than 10% of [14C]DHA remained in endothelial cells of the brain vasculature [40]. Therefore, we considered that [14C]DHA was probably transported across the BBB as NE-DHA. However, we cannot exclude the possibility that [14C]DHA derivatives are also quantified.

Page 34, Lines 590-593: Therefore, it is possible that FABP5 up-regulation in 12- and 24-month-old mice (Fig. 3) may play a crucial role in aged mice for maintaining brain DHA levels and regulating brain uptake of DHA.

Pages 38, Lines 653-656: In conclusion, we report reduced brain uptake of [14C]DHA in middle-aged (12-month-old) and aged (24-month-old) mice. Furthermore, we demonstrated that [14C]DHA is transported across the BBB by a saturable transport system and that MFSD2A partly mediates brain endothelial uptake of [14C]DHA. Finally, we observed…

Other issues to address:

1. Please give evidence (or at least reference) for the purification level of the isolated brain blood vessels (IBV). How can you be sure that the results obtained with IBV are not derived from contaminating non-EC cells in the BV fraction? To be more certain about the conclusions derived from experiments with BV, one should conduct parallel experiments with highly purified BEC for example.

Response: We thank the reviewer for this comment. According to the reviewer’s suggestion, we added a reference and supplemental information regarding the purity of the isolated brain microvessels. We confirmed the purity of the isolated brain microvessels by microscopic observation and western blot analysis as shown below in (Fig. A and B). The isolated brain microvessels showed an enrichment of brain endothelial cell-specific proteins (occludin, claudin-5) and a low abundance of neuronal protein (MAP2) and myelin basic protein (MBP) which are primarily expressed in oligodendrocytes. 

Page 10, Line 162: Brian microvessels were isolated by the modified method of Yousif et al. [24]. Male mice aged…

Pages 11-12, lines 183-185: …-80°C until use. We confirmed that the obtained brain microvessels were enriched with brain endothelial cell-specific proteins (occludin and claudin-5).

Figure A and B. (A) Representative microscopic image of brain microvessels isolated from 2-month-old mice. Scale bar: 50 µm. (B) Western blot images of MAP2, occludin, claudin-5, MBP, and β-actin in the whole brain, brain microvessels isolated from 2-month-old mice, and rat brain endothelial cells isolated from 3- to 4-week-old rats.

2. How do you account for the lack of decrease in BBB permeability to mannitol with age? There is plenty of literature regarding leakiness of the BBB in normal aging, especially in the Hipp (see Zlokovic papers for example). This point needs to be better discussed.

Response: We thank the reviewer for this question. [3H]mannitol is a small hydrophilic molecule (MW 182) that was used as a marker for the paracellular permeability of the BBB. It is regulated by tight junctions and it does not readily cross the BBB (PMID: 33054801). Due to its molecular weight, the BBB permeability to mannitol is higher than that of other paracellular markers with a larger molecular weight (> MW 182), such as sucrose (MW 342). Therefore, mannitol is less sensitive to changes in the paracellular pathway. The lack of decrease in BBB permeability to mannitol at least indicated that the paracellular pathway for mannitol at the BBB was not changed during normal aging. Although we could not detect leakiness of the BBB using [3H]mannitol during normal aging in this study, we were able to find the leakage of a plasma protein fibrinogen at BBB in the hippocampus of 24-month-old mice (Fig. C, D, E). Transcytosis of plasma proteins was upregulated in aged brain microvasculature (PMID: 32612231). Therefore, we thought that age-related BBB disruption, especially transcellular pathways for large molecules, occurred in the brain of 24-month-old mice. Thus, the appearance of age-related changes in BBB permeability depends on the molecular size of a BBB permeability marker. According to the reviewer’s suggestion, we mentioned the leakiness of the BBB in normal aging as follows:

Page 31, Lines 532-535: …aged rats [32]. However, previous works showed the leakiness of human BBB in normal aging [33,34]. These discordances in age-related changes in BBB permeability probably depend on the evaluation methods used, including markers used for BBB permeability. Our findings suggest that…

Figure C, D, E. (C) Representative fluorescent images of fibrinogen (red) and lectin (green) in CA1, CA2, CA3, and DG of the hippocampus. (D and E) Fibrinogen positive areas (D) and intensity € were detected and quantified by fluorescent microscopy in 2- (2M), 12- (12M), and 24-month-old (24M) mice. Data are expressed as fold changes of each corresponding 2M (n = 3). Bars indicate the mean ± standard error of the mean. **P < 0.01, significantly different from each corresponding 2M.

Abbreviations: CA, Cornu Ammonis; DG, dentate gyrus

3. Please show the plateau obtained in the uptake experiments (line 395)

Response: We thank the reviewer for this suggestion. According to the reviewer’s suggestion, we have shown the plateau in the brain uptake of [14C]DHA at 2 min in Fig. 4.

4. Lines: 67-71: This statement regarding ALA is controversial. Whereas dietary maritime DHA supply has been regarded as the main efficient source of brain DHA the ability of its metabolic precursor, terrestrial ALA, to support brain DHA (as well as EPA), has rarely been examined. Recent studies have begun to indicate that ALA enrichment in adult diets and in maternal diets during fetal development and weaning increases brain DHA levels and higher expression of Mfsd2a indicating ALA ability to enable higher DHA levels and improved BBB transport. Please refer to this literature.

Response: We thank the reviewer for the comment. According to the reviewer’s comment, we cited the literature regarding ALA (J Nutr Biochem. 2021) as follows:

Page 5, Lines 68-73: It is controversial to state that the human brain’s capacity to biosynthesize DHA from its precursor, α-linolenic acid (ALA, 18:3n-3), is very low [8], because maternal dietary ALA given during pregnancy-lactation increased brain DHA levels in the offspring [9]. However, plasma DHA, which is obtained directly from dietary intake to maintain DHA levels in the brain, must be transported from the blood to the brain across the BBB [10].

5. Why is sucrose used instead of mannitol? Please discuss the differences between the 2 markers and state the reason for using each marker in the different experiments

Response: We thank the reviewer for this question. As described above, both sucrose and mannitol have commonly been used as markers for paracellular BBB permeability (i.e. BBB integrity) in vivo (PMID: 33444097). A previous study showed that the plasma profiles of mannitol and sucrose were similar, while the brain concentrations and Kin of mannitol were higher than those of sucrose (PMID 33054801).　Because [3H]sucrose and [3H]DHA are not commercially available, we used [3H]mannitol instead of [14C]sucrose for concurrently evaluating BBB integrity and brain [14C]DHA uptake in one mouse. In addition, a co-infusion of an excess amount of DHA and [3H]mannitol resulted in the increased brain/perfusate ratio of [3H]mannitol (Kin for [3H]mannitol was not changed). We assumed that ethanol, as the solvent of DHA, affects it. Therefore, we separately used [14C]sucrose to evaluate BBB integrity for the experiments in the presence of an excess of DHA.

6. In the cellular uptake experiments after transfection, do you account for cellular death? Any other factors that may affect uptake which are not related to mfsd2a levels? Did you try pharmacological inhibition of the transporter?

Response: We thank the reviewer for this question. We did not perform experiments regarding cell viability and cellular dysfunction. However, we confirmed that there were no morphological defects and cellular deaths in siMfsd2a-transfected rat brain endothelial cells. It is possible that transfection of siMfsd2a causes changes in FABP5 expression levels in rat brain endothelial cells which may affect the cellular uptake of DHA. Therefore, further research is needed to clarify the changes in FABP5 expression levels after transfection and its contribution to the decreased uptake of DHA in vitro. Regarding pharmacological inhibition, unfortunately, no pharmacological inhibitors for MFSD2A are available at present. However, we confirmed that the cellular uptake rate of [14C]DHA declined in the presence of unlabeled DHA in vitro　(Fig. F).

Figure F. Cellular uptake rate of [14C]DHA in rat brain endothelial cells in the presence of unlabeled DHA. The cellular uptake rate was calculated from cell / medium ratio at each time point (0.5, 2, 5, 10 min). N=3-6, Data are Means ± SEM. *P < 0.05, significantly different from vehicle.

7. References missing in line 513 regarding DHA and cognitive decline

Response: We appreciate the reviewer’s attention. We added the references regarding DHA and cognitive decline as follows (underlined):

Page 30, Line 510: ~cognitive decline [18,20]. 

Page 30, Line 512: ~across the BBB [7].

8. When counting C14 inside the brain parenchyma, how do you make sure the C14 doesn’t originate from the BV in this tissue without BV depletion? Please describe in more details the brain perfusion technique and why do you think it provides the experimental conditions for the NE-DHA to remain NE. Is it possible that the injected DHA will be metabolized before binding to mfsd2a on the BEC? The same for the in-vitro studies. See PMID: 9886086, as an example: it is a useful reference to understand how a follow up of DHA location may be done.

Response: We thank the reviewer for this question. A previous study indicated that capillary depletion of brain homogenates after in situ brain perfusion (a 40 s perfusion) showed that less than 10% of [14C]DHA remained in the endothelial cells of the brain vasculature and more than 90% of [14C]DHA was detected in brain parenchymal fraction, demonstrating that DHA rapidly crossed the BBB (PMID: 19442696). In addition, the HPLC analysis showed that a main peak containing [14C]DHA was detected at the same retention time in the perfusate and the perfused brain (PMID: 19442696). In the present study, we employed a 0.5- to 1.5-min perfusion. Therefore, we considered that most of the NE-DHA which crossed the BBB remains to be a non-esterified form in brain parenchyma during a 1.5 min perfusion. However, we did not confirm that in this study. We toned down the conclusion in the revised manuscript.

Regarding the metabolism of DHA before binding to Mfsd2a on brain endothelial cells, a previous work by Bernoud et al. using bovine brain endothelial cells showed the passage of non-esterified DHA through the brain endothelial cell monolayer 2 h after adding non-esterified DHA into the upper medium (i.e. the luminal side of brain endothelial cells). Non-esterified DHA taken up by brain endothelial cells was mainly incorporated into phosphatidylcholine during a 2 h incubation (PMID: 9886086). In the present in vitro study, we evaluated the cellular uptake of [14C]DHA during shorter time periods (up to 10 min). Although several enzymes (such as cytochrome P450, lipoxygenase, β-oxidation enzymes, etc.) are associated with metabolizing DHA, these enzymes are present intracellularly. Therefore, we assume that extracellular [14C]DHA is unlikely to be metabolized before binding to Mfsd2a on brain endothelial cells. 

9. Lines 527-529 in discussion: did you test for significance between the regions? Please do, otherwise you cant say that.

Response: We appreciate the reviewer’s suggestion. Two-way ANOVA showed the significant effects of age and regions, but there was no significant interaction between age and regions. We added the results of the two-way ANOVA for the effects of age and regions and revised the sentence regarding the region specificity in the discussion as follows:

Page 17, Lines 290-291: …multiple comparison tests. A two-way ANOVA (age × brain regions) was performed to analyze differences in the brain uptake of [14C]DHA.

Page 19, Lines 313-316: …2-month-old mice. Two-way ANOVA revealed the significant effects of age (F(3,204) = 24.63, P < 0.0001) and regions (F(5,204) = 2.384, P = 0.0396), but no significant interaction between age and regions (F(15,204) = 0.5614, P = 0.9016). 

Page 30, Lines 522-524: There were age-related decreases in the brain uptake of [14C]DHA in specific regions (Fig. 1). These data suggest that the availability of DHA is not influenced by aging in the cortex and cerebellum.

10. The term “vascular space” should be better defined

Response: We appreciate the reviewer’s suggestion. We defined the term “vascular space” as follows:

Pages 30-31, Lines 525-530: Furthermore, we evaluated the brain/perfusate ratio of [3H]mannitol, a small hydrophilic molecule with low paracellular permeability across the BBB. The brain/perfusate ratio of [3H]mannitol reflects the cerebral vascular volume, because [3H]mannitol does not cross the BBB during a short period and remains within the intracerebral vasculature. There were no significant differences in brain uptake of [3H]mannitol among the 2-, 8-, 12-, and 24-month-old groups. Our data is…

11. Lines 581-582: this conclusion doesn’t stem from your results. You didn’t check it.

Response: We appreciate the reviewer’s attention. As the reviewer suggested, we corrected these sentences as follows:

Page 33, Lines 573-577: Taking into account previous studies showing that LPC-DHA is a primary substrate for MFSD2A [16] and FABP5 contributes to the intracellular transport of NE-DHA, which penetrates the luminal membrane of BMECs [14], our results suggest that MFSD2A may also serve as a transporter for extracellular NE-DHA.

12. To strengthen the conclusions derived from the in-vitro studies one should add at least one of the following: competition with cold DHA, pharmacological inhibition, use other substrates which are ligands for other transporters and show they are not affected by the transfection. And again, measurements of radioactivity and not directly DHA using other analytical methods (HPLC and GC) can only provide assumptions and not proofs of what is the exact DHA form in the different stages/regions.

Response: We appreciate the reviewer’s suggestion. As described above, we confirmed that unlabeled DHA decreased the cellular uptake rate of [14C]DHA in rat brain endothelial cells (Fig. F). Considering, as a limitation, that we did not measure DHA using HPLC or LC/MS, we added a limitation of this study and toned down the statements referring to DHA forms in the revised manuscript.

Reviewer #2: 

We appreciate the reviewer’s comments and suggestions, which have enabled us to considerably improve our manuscript. As indicated in the following responses, we have taken all of these comments and suggestions into account in the revised version of our manuscript.

The in situ transcardiac perfusion was developed by William Banks laboratory, and later optimized for fatty acid by Pan et al. (https://pubs.acs.org/doi/10.1021/acs.molpharmaceut.5b00580). Please also include Pan et al as reference here (i.e. current reference 13). Can the authors confirm if they have included BSA in the perfusion fluid, as otherwise 2 mL/min may not give sufficient pressure for proper perfusion. If BSA was not included, please justify if 2 mL/min is appropriate.

Response: To avoid binding [14C]DHA to plasma components (such as albumin), we used BSA-free perfusate in this study according to Pan et al. As the reviewer suggested, we agree that the employed perfusion rate at 2 mL/min does not give sufficient cerebral vascular pressure. However, this perfusion condition resulted in a Brain/perfusate ratio of [14C]DHA ranging from 0.1 to 0.4 mL/g brain with a 1 min perfusion. Although this was one-fourth lower than or almost the same as that reported in the previous work by Pan et al., a brain/perfusate ratio of [14C]sucrose obtained from a 0.5 to 1.5 min perfusion at 2 mL/min is slightly higher than that from a 2 min perfusion at 10 mL/min, which was reported by Pan et al. These differences would be mainly due to insufficient cerebral vascular pressure caused by a lower perfusion rate. Therefore, we recognized that our results of the Kin value and/or brain/perfusate ratio of [14C]DHA may be underestimated or not accurate. In this study, however, we would like to emphasize that the aged brain exhibited a decreased uptake of [14C]DHA compared with the young brain under this perfusion condition even if the flow rate at 2 mL/min did not give sufficient cerebral vascular pressure. Because our syringe pump (Kd Scientific Inc., Holliston, MA, USA; model KDS 100 legacy syringe pump) cannot achieve the flow rate of 10 mL/min, we will further examine whether the higher perfusion rate (10 mL/min) affects the obtained results. 

We cited the reference and mentioned this technical limitation in the Discussion section in the revised manuscript as follows.

Pages 35-36, Lines 606-618: There are technical limitations in this study. First, a transcardiac brain perfusion technique was optimized for fatty acids by Pan et al. [14]. Since the perfusion rate was given at 2 mL/min, without adding BSA in the perfusate, the cerebral vascular pressure was insufficient. The perfusion rate of 10 mL/min is more appropriate. The brain/perfusate ratio of [14C]DHA in the present study is lower than that reported by Pan et al. [14]. Therefore, our results of the Kin value and/or brain/perfusate ratio of [14C]DHA may be underestimated or not accurate due to insufficient cerebral vascular pressure caused by a lower perfusion rate. Of note, we found that the aged brain exhibited a decreased uptake of [14C]DHA compared with the young brain even if the flow rate of 2 mL/min did not give sufficient cerebral vascular pressure. Therefore, further studies are needed to determine whether a higher perfusion rate (10 mL/min) would affect the obtained results. Second,… 

Fig 1 & 2, please express B:P ratio as mL/mg OR mL/g, rather than normalizing all values to that of 2 months old as a %.

Response: We thank the reviewer for this suggestion. According to the reviewer’s suggestion, we have expressed the Brain/Perfusate ratio as mL/g in Fig.1 and 2, without normalizing all values to that of 2 months old as a %. Due to this conversion of the unit, several results of statistical analysis were changed in the revised manuscript. Specifically, brain uptake of [3H]mannitol exerted no significant differences among all age groups (Fig. 2 in the revised manuscript).

Fig 3 - The MW of FABP5 is ~ 15 kDa, but it was labeled as 25 kDa. Is this a mistake? Again, do not normalize to 2 month old.

Response: We appreciate the reviewer’s attention. We detected FABP5 at 15 to 20 kDa. According to the reviewer’s suggestion, we expressed a ratio of FABP5/ total protein in arbitrary units in Fig.3, without normalizing all values to that of 2 months old as a %.

Fig 4 - Please change the unit to mL/g rather than uL/g

Response: We thank the reviewer for this suggestion. According to the reviewer’s suggestion, we have expressed Brain/Perfusate ratio as mL/g in Fig.4.

Line 119-120, please specify isotope for DHA, sucrose in the subheading

Response: We appreciate the reviewer’s suggestion. According to the reviewer’s suggestion, we have specified the isotope for DHA, sucrose, and mannitol in the subheading as follows:

Page 8, Lines 121-122: Measurement of Brain Uptake of [14C]DHA, [3H]Mannitol, and [14C]Sucrose

---

## [Decision Letter · Decision Letter 1]

5 Feb 2023

Aging decreases docosahexaenoic acid transport across the blood-brain barrier in C57BL/6J mice

PONE-D-22-31018R1

Dear Dr. Dohgu,

We’re pleased to inform you that your manuscript has been judged scientifically suitable for publication and will be formally accepted for publication once it meets all outstanding technical requirements.

Kind regards,

Mária A. Deli, M.D., Ph.D.

Academic Editor

PLOS ONE

Additional Editor Comments (optional):

Reviewers' comments:

Reviewer's Responses to Questions

**Comments to the Author**

1. If the authors have adequately addressed your comments raised in a previous round of review and you feel that this manuscript is now acceptable for publication, you may indicate that here to bypass the “Comments to the Author” section, enter your conflict of interest statement in the “Confidential to Editor” section, and submit your "Accept" recommendation.

Reviewer #2: All comments have been addressed

2. Is the manuscript technically sound, and do the data support the conclusions?

Reviewer #2: Yes

3. Has the statistical analysis been performed appropriately and rigorously? 

Reviewer #2: Yes

4. Have the authors made all data underlying the findings in their manuscript fully available?

Reviewer #2: Yes

5. Is the manuscript presented in an intelligible fashion and written in standard English?

Reviewer #2: Yes

6. Review Comments to the Author

Reviewer #2: (No Response)

7. PLOS authors have the option to publish the peer review history of their article (what does this mean?). If published, this will include your full peer review and any attached files.

Reviewer #2: **Yes: **Yijun Pan

---

## [Editor Report · Acceptance letter]

8 Feb 2023

PONE-D-22-31018R1 

Aging decreases docosahexaenoic acid transport across the blood-brain barrier in C57BL/6J mice 

Dear Dr. Dohgu:

I'm pleased to inform you that your manuscript has been deemed suitable for publication in PLOS ONE. Congratulations! Your manuscript is now with our production department. 

Kind regards, 

on behalf of

Prof. Mária A. Deli 

Academic Editor

PLOS ONE